# StaR-KVQA: Structured Reasoning Traces for Implicit-Knowledge Visual Question Answering

## Abstract

Knowledge-based Visual Question Answering (KVQA) requires models to ground entities in images and reason over factual knowledge. We study its implicit-knowledge variant, *IK-KVQA*, where a multimodal large language model (MLLM) is the sole knowledge source, without external retrieval. Yet, MLLMs lack explicit reasoning supervision and produce inconsistent justifications, and generalize poorly after standard supervised fine-tuning (SFT). We present **StaR-KVQA** (*Structured Reasoning Traces for IK-KVQA*), which supervises structured traces—dual symbolic relation paths plus path-grounded natural-language explanations—so that reasoning becomes transparent and verifiable. With one open-source MLLM, StaR-KVQA constructs and selects path-grounded reasoning traces to form a trace-enriched dataset, then fine-tunes via structured self-distillation to align generation with supervision; no external retrievers, verifiers, or curated knowledge bases (KBs) are used, traces are built offline, and inference is a single autoregressive pass. Across benchmarks, StaR-KVQA improves both accuracy and interpretability, achieving up to **+11.3%** higher answer accuracy on OK-VQA over the strongest baseline while exhibiting robust cross-domain generalization.

## 1 Introduction

Knowledge-based Visual Question Answering (KVQA) is a fundamental yet challenging task at the intersection of computer vision, natural language processing, and knowledge reasoning (Wang et al., 2016; Marino et al., 2019; Schwenk et al., 2022a). Unlike conventional VQA, which typically learns a direct mapping from image features to textual answers, KVQA requires models to both *ground entities in the image* and *reason over factual knowledge*. For example, answering the question *"Which breed of dog is this?"* requires recognizing attributes such as color and size from the image, connecting them with prior knowledge about breeds, and producing a faithful explanation that supports the final answer. Such tasks highlight that solving KVQA is not only about accuracy, but also about faithful and interpretable reasoning.

Early approaches often relied on explicit knowledge graphs (KGs) or retrieval modules (Chen et al., 2024). While effective, these pipelines suffer from several limitations: (i) external knowledge requires costly maintenance and is inherently incomplete, (ii) reasoning is fragmented across retrieval, fusion, and prediction modules, reducing transparency, and (iii) errors in recognition or retrieval easily propagate without robust correction. These limitations have motivated the community to explore *implicit-knowledge KVQA (IK-KVQA)*, short for *Implicit-Knowledge Knowledge-based VQA*, where the "K" continues to mark the task as knowledge-based, while multimodal large language models (MLLMs) [1] directly generate answers without retrieval (Yang et al., 2021; Lin et al., 2022), as illustrated in Figure 1. The IK-KVQA setting simplifies system design and removes external dependencies. However, it also poses stricter demands: the model must rely solely on its parameters to ground evidence, recall factual knowledge, and reason. In practice, MLLMs often

---

[1] In the literature, models such as Qwen2.5-VL are also referred to as vision–language models (VLMs). We use the term *MLLMs* in this paper for consistency.

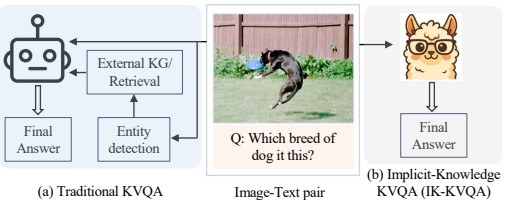

Figure 1: **Comparison between traditional KVQA and implicit-knowledge KVQA (IK-KVQA)**. In contrast, IK-KVQA (*Implicit-Knowledge Knowledge-based VQA*) retains the "K" from KVQA to indicate its knowledge-based nature, but removes external sources: answers are predicted solely from $(I, Q)$ and parametric knowledge $f_\theta(I, Q)$. This stricter setting is more practical and scalable, yet also demands new techniques to ensure faithful and interpretable reasoning.

behave as *black boxes*—producing correct answers but with reasoning that is opaque or inconsistent. The absence of explicit reasoning supervision undermines interpretability and trustworthiness. Concretely, IK-KVQA faces three core challenges: (1) *Lack of explicit supervision*, since models are trained only on final answers while reasoning traces remain hidden; (2) *Weak interpretability*, as predictions are often correct but not accompanied by faithful justifications; and (3) *Limited generalization*, as conventional fine-tuning tends to overfit in-domain and generalizes poorly across domains.

To address these issues, we propose **StaR-KVQA**, short for *Structured Reasoning Traces for Implicit-Knowledge Visual Question Answering*. The acronym **StaR** highlights four key aspects of our design: **S**tructured reasoning paths, **t**races that explicitly record the reasoning process, **a**nswering in the KVQA setting, and **R**easoning supervision that makes the model transparent and verifiable. Together, these elements define our central idea: instead of leaving reasoning implicit, StaR-KVQA supervises both symbolic paths and natural-language explanations as *structured reasoning traces*, enabling the model to deliver not only accurate answers but also faithful reasoning.

StaR-KVQA reuses a single open-source MLLM (e.g., Qwen2.5-VL-7B) to generate dual reasoning paths, compose natural-language explanations, and select the most consistent triplet, producing an augmented dataset with explicit reasoning traces. Fine-tuning on this dataset yields a task model that performs *structured self-distillation*: it learns not only from final answers, but also from intermediate reasoning signals (paths and explanations). This richer supervision provides stronger inductive bias—guiding the model to connect visual cues with factual knowledge step by step—which reduces spurious shortcuts and improves answer accuracy. At inference, the fine-tuned model autoregressively generates reasoning and answers together, providing transparent and verifiable predictions without any external knowledge. Viewed broadly, StaR-KVQA extends self-distillation into the multimodal reasoning regime. Unlike prior text-only formulations that distill final answers, our approach distills *structured intermediate reasoning*, yielding both **stronger accuracy** and **faithful interpretability**.

**Our contributions are threefold.** (i) **Structured Reasoning Traces for IK-KVQA.** We introduce **StaR-KVQA**, which supervises dual symbolic relation paths together with path-grounded natural-language explanations as *structured reasoning traces*, turning reasoning into explicit, transparent, and verifiable signals that function as a principled inductive bias in the IK setting. (ii) **Implementation-friendly single-model pipeline.** We realize a three-stage pipeline—*dual-path planner*, *reasoning composer*, and an *internal selector instantiated with the same model*—to produce high-quality traces and build a trace-enriched dataset for *structure-aware self-distillation*. The system uses a **single** open-source MLLM across all stages, requires no external retrievers/verifiers and no additional trainable modules; trace construction is performed offline, and inference proceeds in a single autoregressive pass without external retrieval. (iii) **Extensive validation and state-of-the-art performance.** Fine-tuning on the trace-enriched data yields consistent gains in both **accuracy** and **interpretability**, up to **+11.3%** on OK-VQA over the strongest baseline, with robust cross-domain generalization.

## 2    RELATED WORK

We review KVQA with retrieval, multimodal/large language models, and self-distillation to position our contributions.

**KVQA with knowledge graphs or retrieval.** Early datasets (FVQA (Wang et al., 2016), OK-VQA (Marino et al., 2019; Schwenk et al., 2022b), KVQA (Shah et al., 2019)) spurred pipelines that integrate explicit KGs or retrievers, e.g., ConceptBERT (Gardères et al., 2020), MAVEx (Wu et al., 2022), and KRISP (Marino et al., 2021). Recent retrieval-augmented systems such as Wiki-LLaVA (Caffagni et al., 2024), RoRA-VLM (Qi et al., 2024), and EchoSight (Yan & Xie, 2024) underscore

the value of external knowledge, yet introduce complexity, error propagation, and maintenance costs, with limited transparency and out-of-domain generalization.

**KVQA with large language models.** To reduce reliance on explicit KGs, LLMs have been used as implicit knowledge engines: PICa (Yang et al., 2022) shows GPT-3 (Brown et al., 2020) can answer knowledge-intensive questions from captions; KAT (Gui et al., 2021) and REVIVE (Lin et al., 2022) add supporting evidence, while MAIL (Dong et al., 2024) and ReflectiVA (Cocchi et al., 2025) explore reflective/adaptive fusion. However, reasoning traces often remain absent or inconsistent, limiting interpretability and verifiability.

**Multimodal large language models.** Recent MLLMs perform end-to-end image–text reasoning via lightweight projections (Liu et al., 2023; 2024), Q-Former (Li et al., 2023; Dai et al., 2023), Perceiver-style modules (Laurençon et al., 2024), or cross-attention as in Flamingo (Alayrac et al., 2022; Awadalla et al., 2023). Training typically combines large-scale caption alignment (Changpinyo et al., 2021; Gadre et al., 2023; Laurençon et al., 2024) with visual instruction tuning (Laurençon et al., 2023). In the IK-KVQA regime, their reasoning remains mostly opaque and weakly supervised.

**Self-distillation and reasoning supervision.** Self-distillation (Zhang et al., 2019; 2021) creates auxiliary supervision from model outputs; SDFT (Yang et al., 2024) rewrites responses to mitigate forgetting, and Wang et al. (2023c) adds structural signals for multi-hop QA. Yet most works distill only final answers, leaving reasoning implicit. We provide *structured reasoning traces*—dual symbolic paths and path-grounded explanations—as explicit supervision for IK-KVQA, enabling single-pass, verifiable reasoning and improved cross-domain robustness via structured self-distillation.

## 3 PRELIMINARIES

We now formalize the IK-KVQA setting, task notation, and assumptions.

**Problem definition**: given an image $I$ and a question $Q$, knowledge-based visual question answering (KVQA) aims to predict an answer $\hat{a} \in \mathcal{A}$:

$$\hat{a} = f(I, Q, K), \tag{1}$$

where $f$ is the answering model and $K$ denotes external knowledge retrieved from a knowledge graph or textual corpus. Traditional KVQA pipelines typically ground entities in the image and query $K$ to supplement factual reasoning before producing the final answer.

**Implicit-knowledge KVQA**: in contrast, we consider the implicit-knowledge setting (IK-KVQA), where $K$ is unavailable. The only information sources are (i) visual evidence from $I$, (ii) linguistic cues from $Q$, and (iii) *parametric knowledge* encoded in the model parameters. Under this setting, the answer is predicted as

$$\hat{a} = f_\theta(I, Q), \tag{2}$$

where $f_\theta$ is trained solely with implicit knowledge. This formulation eliminates external dependencies but leaves reasoning implicit and unverifiable. Our framework addresses this gap by augmenting supervision with explicit reasoning traces (dual paths and natural-language explanations), enabling faithful and interpretable reasoning entirely within the parametric model.

## 4 STRUCTURED REASONING TRACES FOR IK-KVQA

We introduce **StaR-KVQA**[2], which replaces answer-only supervision with *structured reasoning traces*: dual relation paths $(P_t, P_v)$ and a path-grounded explanation $C$. This converts reasoning into explicit, verifiable training signals. The entire pipeline runs within a single open-source $\text{MLLM}_\phi$—producing paths, composing explanations, and internally selecting the best triplet—without external retrievers/verifiers or curated KBs; see Figure 2.

**Design principles.** (i) *Inductive structure:* relation paths stabilize and audit planning; (ii) *Verification loop:* paths→explanation→answer on one track enables consistency checks; (iii) *Single-family learning:* generation and supervision remain style-aligned via self-distillation. Formal notes appear in Appendix A.3.

---

[2]Anonymous code link at `https://anonymous.4open.science/r/kbvqa-110F`

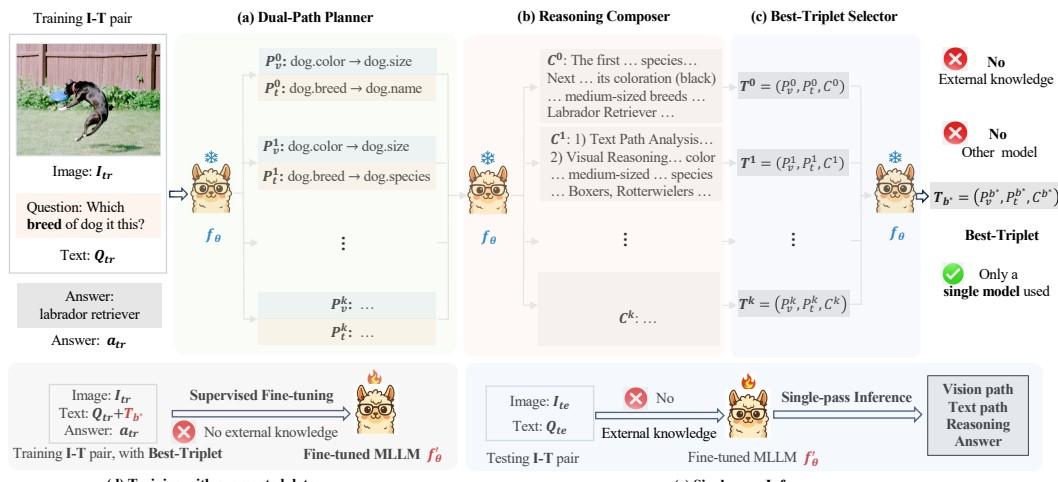

Figure 2: **Overview of StaR-KVQA.** Given a training Image–Text pair, a single $\text{MLLM}_\phi$ generates multiple dual reasoning paths (a) and corresponding explanations (b). A selector (c) identifies the most consistent triplet, which is combined with the ground-truth answer to form reasoning-augmented supervision (d). The fine-tuned model $f'_\theta$ then performs single-pass inference (e), jointly producing reasoning traces and answers without relying on external knowledge.

## 4.1 DUAL-PATH PLANNER

Reasoning in KVQA often requires bridging both linguistic and visual modalities. To explicitly structure this process, we design a **dual-path planner** that produces symbolic *relation paths*. Relation paths capture semantic relations between entities and attributes, and have been widely adopted in knowledge-graph reasoning due to their stability and interpretability (Wang et al., 2021; Xu et al., 2022; Wang et al., 2023a). Unlike dynamically changing entities, relations are more stable, making relation paths reliable surrogates for reasoning plans.

Formally, given an image–question pair $(I, Q)$, the frozen backbone $\text{MLLM}_\phi$ generates $K$ candidate path pairs:

$$\{(P_t^{(k)}, P_v^{(k)})\}_{k=1}^{K} = \text{Planner}_\phi(I, Q), \tag{3}$$

where each $(P_t^{(k)}, P_v^{(k)})$ consists of: (1) a *text path* $P_t^{(k)}$ capturing semantic associations from $Q$ and linguistic priors, and (2) a *vision path* $P_v^{(k)}$ encoding attributes and relations grounded in $I$.

We operationalize *plan-then-solve* ideas for IK-KVQA by planning *internally* over relation paths within a **single-model** setup, without external KGs or retrieval. Compared to prior plan-first prompting (Wang et al., 2023b) and KG path reasoning (Luo et al., 2023), our planner unifies textual priors and visual attributes as dual relation paths inside a **single-model** pipeline, offering multiple candidate routes before explanations are generated. This internal planning plays the role of an *inductive bias*: it narrows the search space while keeping the plan auditable.

For example, consider the question `"Which breed of dog is this?"` with image $I$. One candidate might be

$$P_v^{(k)}: \text{dog.color} \rightarrow \text{dog.size}, \qquad P_t^{(k)}: \text{dog.breed} \rightarrow \text{dog.name}.$$

These complementary relation paths define plausible reasoning trajectories that connect visual cues with semantic priors, thereby reducing spurious shortcuts and enhancing both interpretability and accuracy.

## 4.2 REASONING COMPOSER

Given a dual-path pair $(P_t^k, P_v^k)$, the **reasoning composer** turns abstract plans into natural-language *reasoning content* $C^k$ using the **same single-model** backbone:

$$C^k = \text{Compose}_\phi(I, Q, P_t^k, P_v^k). \tag{4}$$

We follow the evidence that explanations can act as supervision: VQA-NLE-style rationales improve answer quality and interpretability (Suo et al., 2023; Irawan et al., 2024; Xie et al., 2024); chain-of-thought prompts in ScienceQA induce more structured reasoning (Lu et al., 2022; Zhang et al., 2023b); and introducing explicit clues or enforcing explanation–answer agreement (DCLUB, MCLE) reduces shortcutting and inconsistency (Fu et al., 2023; Lai et al., 2023).

Our composer instantiates these insights *within the IK setting* by *binding* the rationale to the proposed paths: mentions in $C^k$ cite relations/attributes present in $P_v^k$ and semantic hops in $P_t^k$, discouraging free-form but ungrounded narratives. This binding keeps explanations concise, verifiable against the paths, and aligned with the final answer; practically, it converts interpretability into a supervision signal that the **single-model** system can learn.

## 4.3 BEST-TRIPLET SELECTOR

Not all generated triplets $(P_t^k, P_v^k, C^k)$ are reliable, and directly using them may introduce noisy or inconsistent supervision. To address this, we introduce a **best-triplet selector** that filters candidates during the *data augmentation stage*, where raw dual paths and reasoning contents are expanded into training signals.

Concretely, the selector is instantiated as an *LLM-as-a-judge* within the same **single-model** setup, reusing the backbone $\text{MLLM}_\phi$. Given $(I, Q)$ and a set of candidates $\{(P_t^k, P_v^k, C^k)\}_{k=1}^K$, we prompt $\text{MLLM}_\phi$ to rank triplets according to three criteria: (i) *path–answer consistency* (the answer follows from the explanation and paths), (ii) *internal coherence and conciseness*, and (iii) *faithfulness* (explicitly citing elements from $P_t^k/P_v^k$). Formally, the preference score is denoted by $s_\phi$:

$$b^* = \arg\max_b s_\phi(I, Q, P_t^b, P_v^b, C^b), \quad T_{b^*} = (P_t^{b^*}, P_v^{b^*}, C^{b^*}). \tag{5}$$

This step introduces *no additional trainable parameters*. By reusing the same backbone that generated the candidates, the augmented dataset $\mathcal{D}_{\text{aug}}$ stays stylistically aligned with the model's own traces and favors reasoning explanations that remain stable under light paraphrasing. In practice, this design yields higher-quality supervision while keeping the pipeline lightweight and parametric-only. All the related prompts are in Appendix A.6.

**Why single-model trace construction (vs. extra modules)?** We deliberately avoid additional verifiers or retrievers for three reasons aligned with the IK setting: (i) *Homogeneous generation–learning (structured self-distillation):* planning, composing, and selecting are all performed by the same family, so the fine-tuned student $f_\theta$ learns from traces already in the generator's style, mitigating supervision–generation mismatch and catastrophic forgetting (Yang et al., 2024); (ii) *Test-time simplicity:* selection is performed only offline during augmentation, leaving inference as a single autoregressive pass; (iii) *IK compliance:* the design remains fully parametric, requiring no external knowledge or extra modules, which makes the reasoning process easier to audit as the model emits paths, explanations, and answers in one decoding stream.

## 4.4 TRAINING WITH AUGMENTED DATA

Let the training split be $\{(I_{tr}^i, Q_{tr}^i, a_{tr}^i)\}_{i=1}^N$, where $I_{tr}^i$ is the image, $Q_{tr}^i$ the question, and $a_{tr}^i$ the ground-truth answer for the $i$-th instance. For each pair $(I_{tr}^i, Q_{tr}^i)$, the planner and composer generate multiple candidate triplets, and the selector chooses the best one $T_{b^*}^i = (P_t^{b^*}, P_v^{b^*}, C^{b^*})$. We then construct the **augmented training set** by attaching the selected reasoning trace to each instance:

$$\mathcal{D}_{\text{aug}} = \{(I_{tr}^i, Q_{tr}^i, T_{b^*}^i, a_{tr}^i)\}_{i=1}^N. \tag{6}$$

This enriched dataset provides explicit supervision over both answers and reasoning traces. The base model $f_\theta$ is then fine-tuned on $\mathcal{D}_{\text{aug}}$ using a standard token-level cross-entropy loss:

$$\mathcal{L}_{\text{SFT}}(\theta; \mathcal{D}_{\text{aug}}) = - \sum_{(I,Q,T,a) \in \mathcal{D}_{\text{aug}}} \log p_\theta(T, a \mid I, Q), \tag{7}$$

where the target sequence concatenates the reasoning paths $P_t, P_v$, reasoning content $C$, and the final answer $a$. This objective encourages the fine-tuned model $f'_\theta$ to jointly generate structured reasoning traces and correct answers in a single, coherent output.

### 4.5 Single-pass Inference

At test time, given $(I_{te}, Q_{te})$, the fine-tuned model $f'_\theta$ performs a *single* autoregressive decode that jointly emits dual paths, a path-grounded explanation, and the final answer:

$$f'_\theta(I_{te}, Q_{te}) = (\hat{P}_t, \hat{P}_v, \hat{C}, \hat{a}). \tag{8}$$

No selector or auxiliary module is invoked at inference. This design preserves the verification loop (paths→explanation→answer) and exposes a complete trace for auditing, with no external retrieval.

## 5 Experiments

Table 1: Performance comparison on OK-VQA.

| Method | Model Inputs | External Knowledge | Acc. (%) |
|---|---|---|---|
| Q Only | Question + Image | - | 14.93 |
| **KVQA with Knowledge Graphs and Retrieval** | | | |
| BAN | Question + Image | - | 25.17 |
| BAN +AN | Question + Image | Wikipedia | 25.61 |
| MUTAN | Question + Image | - | 26.41 |
| MUTAN +AN | Question + Image | Wikipedia | 27.84 |
| ConceptBERT | Question + Image | ConceptNet | 33.66 |
| HCNMN | Question + Image | WordNet | 36.74 |
| Krisp | Question + Image | Wikipedia + ConceptNet | 38.90 |
| MAVEx | Question + Image | Wikipedia + ConceptNet + Google Images | 41.37 |
| VLC-BERT | Question + Image | COMET + ConceptNet | 43.14 |
| MCAN | Question + Image | - | 44.65 |
| **KVQA with Large Language Models** | | | |
| PICA-Base | Question + Caption + Object Tags | Frozen GPT-3 (175B) | 43.30 |
| Pica-Full | Question + Caption + Object Tags | Frozen GPT-3 (175B) | 48.00 |
| KAT (Single) | Question + Caption + Object Tags | Frozen GPT-3 (175B) + Wikidata | 53.09 |
| KAT (Ensemble) | Question + Caption + Object Tags | Frozen GPT-3 (175B) + Wikidata | 54.41 |
| REVIVE | Question + Caption + Region Tags | Frozen GPT-3 (175B) + Wikidata | 53.83 |
| MAIL | Question + Image | Frozen MiniGPT-4 (7B) + ConceptNet | 56.69 |
| **IK-KVQA with Multimodal Large Language Models** | | | |
| Qwen2.5-VL-7B | Question + Image | Qwen2.5-VL-7B | 75.74 |
| Llama-3.2-11B-Vision | Question + Image | Llama-3.2-11B-Vision | 67.84 |
| Gemma-3-12B | Question + Image | Gemma-3-12B | 71.40 |
| Gemma-3-27B | Question + Image | Gemma-3-27B | 79.34 |
| Qwen2.5-VL-72B | Question + Image | Qwen2.5-VL-72B | 80.75 |
| InternVL3-78B | Question + Image | InternVL3-78B | 67.61 |
| GPT-4o | Question + Image | GPT-4o | 77.86 |
| Gemini 2.5 Flash | Question + Image | Gemini 2.5 Flash | 79.97 |
| Gemini 2.5 Pro | Question + Image | Gemini 2.5 Pro | 80.53 |
| SFT | Question + Image | Fine-tuned Qwen2.5-VL-7B | 76.36 |
| COT | Question + Image | Qwen2.5-VL-7B | 76.88 |
| LLaVA-CoT | Question + Image | Fine-tuned Llama-3.2-11B-Vision | 76.57 |
| M2-Reasoning | Question + Image | M2-Reasoning-7B | 78.63 |
| SDFT | Question + Image | Fine-tuned Qwen2.5-VL-7B | 82.56 |
| StaR-KVQA$_{Qwen}$ | Question + Image | Fine-tuned Qwen2.5-VL-7B | **91.51** |
| StaR-KVQA$_{Llama}$ | Question + Image | Fine-tuned Llama-3.2-11B-Vision | **90.01** |
| StaR-KVQA$_{Gemma}$ | Question + Image | Fine-tuned Gemma-3-12B | **91.90** |

In this section, we conduct comprehensive experiments to evaluate the effectiveness of our proposed framework, **StaR-KVQA**. Our experiments address the following research questions: **RQ1 (Main Results):** Does StaR-KVQA improve answer accuracy compared with state-of-the-art baselines under the implicit-knowledge setting? **RQ2 (Ablation Studies & Hyperparameters):** How do structured reasoning traces—dual paths and natural-language explanations—contribute to performance, and how sensitive is the framework to the number of candidate paths $K$? **RQ3 (Cross-domain Generalization):** Can StaR-KVQA maintain robustness when transferring across datasets? **RQ4 (Case Study):** Do the generated reasoning traces enhance interpretability and provide faithful justifications?

## 5.1 EXPERIMENTAL SETUP

**Datasets.** In line with recent advances in the field (Marino et al., 2019; Yang et al., 2021; Gui et al., 2021; Wu et al., 2022; Lin et al., 2022), we performed our primary validation on the OK-VQA dataset. Comprising 14,055 image-question pairs, this benchmark is currently the most demanding in the domain. Furthermore, to establish the broader applicability of our model, we performed supplementary experiments on FVQA (Wang et al., 2016), the original dataset that initiated the exploration of KVQA.

**Baselines.** We employ three categories of baselines for comparison, with details provided in Appendix A.1. (i) **KVQA with Knowledge Graphs and Retrieval**, which construct diverse multimodal learning frameworks to perform final reasoning on the given questions. (ii) **KVQA with Large Language Models**, which integrate large language models either to directly predict the answer or to generate relevant supporting evidence.(iii) **IK-KVQA with Multimodal Large Language Models**, an MLLM directly generates the answer. Note: all the MLLMs here are **instruction-tuned** versions. And they are evaluated under a uniform protocol: fixed seed 42, default decoding, no CoT, and inputs limited to (image, question). Implementation details is in Appendix A.1.1. To evaluate our model, we utilized the direct answer setting, wherein the model generates open-ended text. The responses were then scored according to the standard VQA evaluation from (Agrawal et al., 2015), with details in Appendix A.2.

## 5.2 MAIN RESULTS

Table 2: Performance comparison of IK-KVQA with MLLMs approaches on **FVQA**.

| Method | Acc. (%) |
|---|---|
| Qwen2.5-VL-7B | 71.61 |
| Llama-3.2-11B-Vision | 66.09 |
| Gemma-3-12B | 70.64 |
| Gemma-3-27B | 76.82 |
| Qwen2.5-VL-72B | 75.95 |
| InternVL3-78B | 70.99 |
| GPT-4o | 72.36 |
| Gemini 2.5 Flash | 74.51 |
| Gemini 2.5 Pro | 73.39 |
| SFT | 73.91 |
| COT | 74.66 |
| LLaVA-CoT | 78.45 |
| M2-Reasoning | 72.53 |
| SDFT | 75.54 |
| StaR-KVQA$_{Qwen}$ | **82.82** |
| StaR-KVQA$_{Llama}$ | **80.19** |
| StaR-KVQA$_{Gemma}$ | **81.20** |

To answer RQ1, we summarize the comparisons with representative baselines in Table 1 and Table 2. Several key observations can be made: (i) **MLLMs as strong backbones.** Methods based on state-of-the-art multimodal large language models (MLLMs) achieve the strongest overall performance, even without explicit external knowledge. This confirms that the parametric knowledge encoded in large-scale pretraining is already highly effective for KVQA, while also being easier to use compared with retrieval- or KG-based approaches. (ii) **StaR-KVQA achieves the best results.** Among the MLLM-based methods, our proposed *reasoning-augmented framework* consistently delivers the best performance. On OK-VQA, it surpasses the strongest baseline by an impressive **+11.3%**, clearly demonstrating the effectiveness of augmenting training with *structured reasoning traces*. (iii) **Closed-source models are strong but surpassed.** As expected, closed-source commercial systems achieve competitive results, but still fall short of our approach. Notably, StaR-KVQA outperforms **Gemini 2.5 Pro**, one of the most advanced multimodal reasoning models to date. (4) **Self-distillation is strong but limited.** We further introduce **Self-Distillation Fine-Tuning (SDFT)** (Yang et al., 2024), which rewrites task responses into the model's own style for fine-tuning. With Qwen2.5-VL-7B as the backbone, SDFT delivers remarkable performance—exceeding Gemini 2.5

Table 3: Ablation studies.

| Methods | Vision Path | Text Path | Reasoning Composer | Best-Triplet Selector | OK-VQA | | | FVQA | | |
|---|---|---|---|---|---|---|---|---|---|---|
| | | | | | Qwen | Llama | Gemma | Qwen | Llama | Gemma |
| **Variant 1** | × | × | ✓ | ✓ | 87.47 | 72.57 | 89.09 | 76.31 | 76.31 | 79.14 |
| **Variant 2** | ✓ | ✓ | × | ✓ | 87.53 | 86.00 | 88.26 | 76.22 | 76.05 | 73.34 |
| **Variant 3** | ✓ | × | ✓ | ✓ | 83.66 | 72.77 | 87.84 | 76.91 | 56.91 | 79.66 |
| **Variant 4** | × | ✓ | ✓ | ✓ | **92.65** | 70.01 | 86.92 | 74.42 | 64.81 | 78.20 |
| **Variant 5** | ✓ | ✓ | ✓ | × | 91.76 | 72.17 | **91.94** | **84.55** | 49.18 | **83.18** |
| **StaR-KVQA** | ✓ | ✓ | ✓ | ✓ | 91.51 | **90.01** | 91.90 | 82.82 | **80.19** | 81.20 |

Pro by over 2% on OK-VQA and ranking just below our method. This highlights the strength of self-distillation for IK-KVQA. Yet StaR-KVQA goes further: by supervising both symbolic paths and natural-language explanations as *structured reasoning traces*, it combines SDFT's accuracy gains with faithful, interpretable reasoning, closing the transparency gap left by SDFT. In summary, StaR-KVQA not only surpasses strong open-source and closed-source baselines but also sets a new state of the art in IK-KVQA, combining superior accuracy with transparent reasoning.

## 5.3 Ablation Studies & Hyperparameters

**Ablation Studies.** To answer RQ2, we conduct a series of ablations to examine the role of each component in our reasoning-augmented framework (Table 3). For each variant, we rebuild the augmented training data and retrain the model, ensuring that the reported performance fully reflects the absence of the removed component. The results show that removing either the dual paths ( **no paths**) or the explanations ( **no content**) leads to clear accuracy drops, confirming that symbolic paths and natural language reasoning provide complementary supervision. Restricting the framework to a single modality (**text-only** or **vision-only**) further degrades performance, underscoring the need to integrate textual priors with visual grounding. Finally, replacing the best-triplet selector with random selection (**no-selector**) yields mixed outcomes: it slightly improves Qwen and Gemma on some datasets but severely harms Llama, indicating that the selector is crucial for robustness across diverse backbones, even if random choice occasionally preserves strong candidates. Overall, these ablations verify that *dual paths, reasoning content, and the selector* are indispensable, and their synergy explains why our full **StaR-KVQA** model consistently delivers strong and balanced performance across benchmarks.

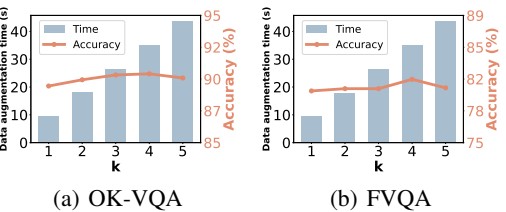

(a) OK-VQA      (b) FVQA

Figure 3: $K$, the number of candidate paths.

**Hyperparameters.** To investigate how sensitive our framework is to the number of candidate paths $K$, we conduct experiments using Qwen2.5-VL-7B as the backbone. For each value of $K$, we report both the final answer accuracy and the average time cost of running the full StaR-KVQA data augmentation pipeline per training example. As shown in Figure 3, several observations emerge: increasing $K$ initially improves performance by providing richer reasoning options, but when $K$ is too large (e.g., $K = 5$), accuracy drops due to overly long contexts that hinder the selector. Overall, the framework is not highly sensitive to $K$, and since augmentation time grows almost linearly with $K$ while gains quickly saturate, a moderate choice such as $K = 3$ achieves the best balance of efficiency and effectiveness.

## 5.4 Cross-domain Generalization

The ability to generalize to out-of-distribution (OOD) data is crucial in real-world applications. To assess this, we evaluate both *in-domain* and *cross-domain* generalization across OK-VQA and FVQA. We consider three model variants: **Frozen** (frozen backbone without training), **SFT** (standard supervised fine-tuning), and our proposed StaR-KVQA, using three MLLM backbones.

**In-domain generalization:** when training and testing are conducted within the same dataset (OK-VQA or FVQA), both SFT and our method yield substantial gains over the Frozen model (left half of Table 4). This confirms that fine-tuning facilitates effective domain adaptation under in-domain

Table 4: Cross-domain generalization.

| | In-domain generalization | | Cross-domain generalization | |
|---|---|---|---|---|
| Source (Tuning) | OK-VQA | FVQA | OK-VQA | FVQA |
| Target (Testing) | OK-VQA | FVQA | FVQA | OK-VQA |
| Frozen$_{Qwen}$ | 75.74 | 71.61 | 71.61 | 75.74 |
| SFT$_{Qwen}$ | 76.36 (+0.62) | 73.91 (+2.30) | 64.77 (-6.84) | 67.50 (-8.24) |
| StaR-KVQA$_{Qwen}$ | 91.51 (+15.77) | **82.82** (+11.21) | 82.09 (+10.48) | **85.45** (+9.71) |
| Frozen$_{Llama}$ | 67.84 | 66.09 | 66.09 | 67.84 |
| SFT$_{Llama}$ | 75.30 (+7.46) | 74.68 (+8.59) | 63.45 (-2.64) | 64.19 (-3.65) |
| StaR-KVQA$_{Llama}$ | 90.01 (+22.17) | **80.19** (+14.10) | 80.09 (+14.00) | **79.59** (+11.75) |
| Frozen$_{Gemma}$ | 71.40 | 70.64 | 70.64 | 71.40 |
| SFT$_{Gemma}$ | 74.45 (+3.05) | 73.73 (+3.09) | 66.83 (-3.81) | 63.91 (-7.49) |
| StaR-KVQA$_{Gemma}$ | 91.90 (+20.50) | **81.20** (+10.56) | 81.20 (+10.56) | **83.43** (+12.03) |

conditions, with our framework further enhancing performance by explicitly supervising reasoning. **Cross-domain generalization:** we then examine transfer across datasets, including both directions: OK-VQA → FVQA and FVQA → OK-VQA. This setting introduces significant ubstantial dataset distribution shift. As shown in the right half of Table 4, SFT exhibits severe degradation—sometimes even underperforming the Frozen baseline—highlighting its vulnerability to catastrophic forgetting and limited generalization. In contrast, our reasoning-augmented framework consistently avoids such degradation and even improves performance on the unseen domain, demonstrating strong robustness against forgetting and superior cross-domain generalization.

## 5.5 QUALITATIVE CASE STUDY

To complement the quantitative results, Table. 5 presents a held-out IK-KVQA example at inference time (*"Can you name the place where this sport is played?"*). The baseline (Qwen2.5-VL-7B) offers a plausible free-form description but does not commit to a canonical venue and drifts toward generic phrases, reflecting the ambiguity in the annotator answers. In contrast, StaR-KVQA produces a complete trace in a single pass—dual relation paths and a path-grounded explanation—that ties the venue prediction to concrete visual and semantic cues (e.g., `sports.name→sports.location` on the text path), yielding a venue aligned with the answer space and making any label bias explicit. This case exemplifies how structured, verifiable traces reduce "right answer, wrong reason" behavior and provide an auditable rationale. Notably, **Gemini 2.5 Pro** also failed to produce the correct venue, underscoring the difficulty of this example even for strong closed-source multimodal reasoning models. Additional examples are provided in the Appendix. A.7.

## 6 CONCLUSION

We have presented **StaR-KVQA**, a reasoning-augmented framework for implicit-knowledge visual question answering (IK-KVQA). By supervising both symbolic paths and natural-language explanations as *structured reasoning traces*, our method transforms reasoning from an implicit by-product into explicit and verifiable steps. Through structured self-distillation, StaR-KVQAachieves state-of-the-art results across multiple benchmarks and backbones, surpassing even advanced closed-source models such as Gemini 2.5 Pro. These gains highlight the effectiveness of structured reasoning supervision for improving both accuracy and interpretability in IK-KVQA. Despite these advances, our framework inherits the risk of *hallucination* from its MLLM backbone, where reasoning traces or answers may still appear plausible but factually incorrect. While structured traces provide a transparent basis for detecting such errors, fully mitigating hallucination remains an open challenge. Future work could explore integrating retrieval-based verification, external consistency checkers, or human-in-the-loop supervision to further strengthen factual grounding. Overall, StaR-KVQArepresents a step toward unifying strong performance with faithful reasoning, and we believe it opens promising directions for advancing transparent multimodal question answering.

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

# A   APPENDIX

## A.1   BASELINES

**KVQA with Knowledge Graphs and Retrieval.** We select representative state-of-the-art approaches, including direct question-only answering (Q Only) (Marino et al., 2019), BAN (Kim et al., 2018), MUTAN (Ben-younes et al., 2017), ConceptBERT (Gardères et al., 2020), KRISP (Marino et al., 2021), MAVEx (Wu et al., 2022), VLCBERT (Ravi et al., 2022), HCNMN (Zhang et al., 2023a), and MCAN (Yu et al., 2019). Since BAN and MUTAN are limited to learning unimodal visual features, we enhance them with ArticleNet (AN) (Marino et al., 2019), which retrieves relevant information from Wikipedia based on the given question–image pair to support external knowledge reasoning. These enhanced versions are referred to as "BAN + AN" and "MUTAN + AN" (Marino et al., 2019).

**KVQA with Large Language Models.** We employ PICa (Yang et al., 2021), KAT (Gui et al., 2021), and REVIVE (Lin et al., 2022). The results of *KVQA with Knowledge Graphs and Retrieval* as well as *KVQA with Large Language Models* are from prior work (Dong et al., 2024), where the exact same experimental setup and evaluation protocols are adopted.

**IK-KVQA with Multimodal Large Language Models.** We employed three types of Multimodal Large Language Models (MLLMs):

- **Advanced open-source MLLMs**: Including three regular-sized models: Qwen2.5-VL-7B (Bai et al., 2025), Llama-3.2-11B-Vision (Dubey et al., 2024), and Gemma-3-12B (Kamath et al., 2025); as well as three larger and more advanced models: Gemma-3-27B (Kamath et al., 2025), Qwen2.5-VL-72B (Bai et al., 2025), and InternVL3-78B (Zhu et al., 2025). All of them are **instruction-tuned versions**.

- **Proprietary state-of-the-art MLLMs**: Including two of Google's most advanced models, Gemini 2.5 Flash and Gemini 2.5 Pro (Comanici et al., 2025), as well as OpenAI's flagship multimodal model, GPT-4o (Hurst et al., 2024). Both Gemini 2.5 Flash and Gemini 2.5 Pro perform inference in the Dynamic Thinking mode.

- **Augmented MLLMs**:
  - **Supervised fine-tuning (SFT)** (Ouyang et al., 2022) is a crucial process that trains a pre-trained MLLM on a high-quality dataset of instructions and responses, making it more effective at following specific commands and performing user-facing tasks. The MLLM backbone is Qwen2.5-VL-7B.
  - **Chain of Thought (CoT)** (Wei et al., 2022) is a prompting technique that improves the reasoning abilities of large language models by guiding them to break down a complex problem into a series of intermediate steps before providing a final answer. The MLLM backbone is Qwen2.5-VL-7B.
  - **LLaVA-CoT** (Xu et al., 2024), a new multimodal model that uses a chain-of-thought method to improve vision-language models' ability to reason step-by-step.
  - **M2-Reasoning** (7B) (AI et al., 2025) is a multimodal large language model (MLLM) that achieves state-of-the-art (SOTA) performance in both general and spatial reasoning by using a high-quality data pipeline and a dynamic multi-task training strategy.
  - **Self-Distillation Fine-Tuning (SDFT)** (Yang et al., 2024) rewrites task responses into its own style and fine-tunes on them to reduce distribution shift and forgetting. The MLLM backbone is Qwen2.5-VL-7B.

### A.1.1   IMPLEMENTATION DETAILS.

Our approach StaR-KVQA has been implemented using PyTorch 2.7.0 as well as Python 3.10, and all experiments have been conducted on the NVIDIA L20 GPU. During training, the batch size (with accumulation) is set to 16, the learning rate is $1e-4$, the LoRA rank is 32, the LoRA alpha is 64, the traininig epoch is 3. In the OK-VQA dataset, $K$ is set as 3, and in the FVQA dataset, $K$ is set as 4.

We adhere to the established evaluation setting and fix the random seed to 42 throughout data loading, parameter initialization, and decoding. Consistent with prior work (Dong et al., 2024), we report single-run results in the main tables to maintain strict comparability with published baselines. We did

not sweep over seeds or report standard deviations; we view multi-seed evaluation as complementary and leave it to future extensions or large-scale replication studies.

To ensure a level playing field across closed- and open-source models, we (i) supply only the image and the question as inputs, without chain-of-thought or auxiliary prompts; and (ii) adopt each model's *default* inference hyperparameters (decoding temperature and maximum generation length), avoiding any model-specific tuning. This protocol matches the default settings recommended by the model providers and prevents gains from hyperparameter overfitting.

## A.2 METRIC

For the open-ended task, *i.e.*, direct answer (DA) setting, we evaluate generated answers using the following accuracy definition:

$$\text{Accuracy} = \min\left(\frac{\#\text{humans that provided that answer}}{3}, 1\right) \tag{9}$$

*i.e.*, an answer is considered fully correct (100% accuracy) if it matches the responses of at least three annotators. Before comparison, all responses are normalized by lowercase, converting numbers to digits, and removing punctuation and articles. We deliberately avoid soft similarity measures such as Word2Vec (Mikolov et al., 2013), which may incorrectly cluster semantically distinct words (e.g., "left" vs. "right"). Likewise, we exclude machine translation metrics such as BLEU and ROUGE, as they are mainly suited for multi-word sentence evaluation rather than short answers typically found in VQA.

## A.3 THEORETICAL NOTES FOR STAR-KVQA

This appendix offers compact analyses that formalize how (i) typed, path-grounded traces (planner + *reasoning composer*), (ii) the single-model selector, and (iii) single-model self-distillation contribute to StaR-KVQA. The statements are backbone-agnostic and match the components introduced in Sec. 4.

### A.3.1 NOTATION AND STANDING ASSUMPTIONS

Let $(I, Q, a^\star) \sim \mathcal{D}$ denote image, question, and ground-truth answer. A *trace* is $T = (P_t, P_v, C)$. Our model with parameters $\theta$ induces

$$p_\theta(T, a \mid I, Q) = p_\theta(P_t, P_v \mid I, Q)\, p_\theta(C \mid I, Q, P_t, P_v)\, p_\theta(a \mid I, Q, T). \tag{10}$$

We reuse two structural predicates from Sec. 4.2:

$$\text{Cover}(C; P_t, P_v) \geq \kappa, \qquad \text{Vis}(C; I) \geq \rho, \tag{11}$$

encoding path–sentence coverage and visual attestability. Define the feasible set $\mathcal{T}_{\kappa,\rho} = \{T : \text{Cover} \geq \kappa, \text{Vis} \geq \rho\}$.

### A.3.2 GENERALIZATION BENEFIT FROM TYPED AND VERIFIABLE TRACES

We compare an *answer-only* class with a *trace-constrained* class that must produce $T \in \mathcal{T}_{\kappa,\rho}$ alongside $a$.

**Hypothesis classes.** Let $\mathcal{H}_{\text{ans}} = \{h : (I, Q) \mapsto a\}$ and

$$\mathcal{H}_{\text{trace}} = \{h : (I, Q) \mapsto (T, a) \text{ s.t. } T \in \mathcal{T}_{\kappa,\rho}\}. \tag{12}$$

Both are realized by the *same* architecture but trained with different supervision.

**Theorem 1** (Rademacher shrinkage via verifiable structure)**.** *Assume bounded losses* $\ell(a, a^\star) \in [0, 1]$ *and* $\ell_{\text{trace}}(T, a; a^\star) \in [0, 1]$ *with* $\ell_{\text{trace}}(T, a; a^\star) \geq \ell(a, a^\star)$ *and equality whenever* $T \in \mathcal{T}_{\kappa,\rho}$*. Then for any sample size* $N$ *and* $\delta \in (0, 1)$*, with probability at least* $1 - \delta$*,*

$$\mathcal{R}_{\mathcal{D}}(h_{\text{trace}}) \leq \widehat{\mathcal{R}}_N(h_{\text{trace}}) + 2\,\Re_N(\mathcal{H}_{\text{trace}}) + \sqrt{\frac{\ln(1/\delta)}{2N}}, \tag{13}$$

*and moreover* $\Re_N(\mathcal{H}_{\text{trace}}) \leq \Re_N(\mathcal{H}_{\text{ans}}) \cdot \sqrt{\Pi(\mathcal{T}_{\kappa,\rho})/\Pi(\mathcal{T})}$*, where* $\Re_N(\cdot)$ *is the empirical Rademacher complexity and* $\Pi(\cdot)$ *the growth function.*

*Intuition.* Enforcing typed, verifiable traces prunes implausible labelings (fewer admissible traces per example), which lowers the effective complexity term and tightens the bound. *Practical takeaway.* Structure acts as an inductive bias without changing the backbone.

### A.3.3 Selector as Maximum Likelihood under a Consistency-Noise Model

Our best-triplet selector uses the *single-model* setup to score candidates. The score can be interpreted as a log-likelihood under a simple noise model.

**Model.** For candidate $b$, define binary indicators $Y_b^{(\text{ans})}, Y_b^{(\text{ent})}, Y_b^{(\text{align})}, Y_b^{(\text{coh})} \in \{0,1\}$ for answer correctness, explanation⇒answer entailment, path→explanation alignment, and explanation coherence. Assume conditional independence given a latent quality $q_b$:

$$\Pr(Y_b^{(j)} = 1 \mid q_b) = \sigma(w_j q_b), \qquad j \in \{\text{ans, ent, align, coh}\}, \tag{14}$$

with logistic $\sigma$ and weights $w_j > 0$. Let $\hat{y}_b^{(j)} \in [0,1]$ be soft proxies estimated by the model; the log-likelihood is $\log L_b(q_b) = \sum_j \hat{y}_b^{(j)} \log \sigma(w_j q_b) + (1 - \hat{y}_b^{(j)}) \log(1 - \sigma(w_j q_b))$.

**Proposition 2** (Selector equals MLE/MAP ranking). *The maximizer $\hat{q}_b = \arg\max_q \log L_b(q)$ is monotone in $s_\phi(b) := \sum_j w_j(2\hat{y}_b^{(j)} - 1)$. Therefore selecting $b^\star = \arg\max_b s_\phi(b)$ agrees with MLE (and with MAP under any log-concave prior).*

*Intuition.* The weighted consistency cues act like independent "votes." A larger weighted sum implies a larger MLE quality and thus a higher rank. *Practical takeaway.* Our LLM-as-a-judge ranking matches likelihood-based selection under a reasonable noise model.

### A.3.4 Single-Model Self-Distillation Reduces Supervision–Generation Shift

Let $P$ be the generator distribution over traces (from $\text{MLLM}_\phi$) and $Q_\theta$ the student's distribution after fine-tuning. Let $\mathcal{L} \in [0,1]$ be a bounded loss on completions.

**Lemma 1** (Risk gap upper bounded by divergence). *For any $(I, Q)$,*

$$\left| \mathbb{E}_{T \sim P} \mathcal{L}(T) - \mathbb{E}_{T \sim Q_\theta} \mathcal{L}(T) \right| \leq \sqrt{2\,\text{KL}(P \| Q_\theta)}. \tag{15}$$

*Proof.* By total variation (TV) and Pinsker's inequality: $\left| \mathbb{E}_P f - \mathbb{E}_Q f \right| \leq 2\,\text{TV}(P, Q)$ for $f \in [0,1]$, and $\text{TV}(P, Q) \leq \sqrt{\frac{1}{2}\text{KL}(P\|Q)}$. Combining gives the stated bound. $\square$

**Theorem 3** (Self-distillation alignment). *If fine-tuning reduces $\text{KL}(P\|Q_\theta)$ on augmented traces (i.e., the student learns from traces in the generator's style), the supervision–generation risk gap is $O(\sqrt{\text{KL}(P\|Q_\theta)})$ by Lemma 1. Using a* single-model *setup (shared format/tokenization) typically attains a smaller KL than heterogeneous teachers.*

*Intuition.* Learning from "in-style" traces narrows the distribution gap, which directly controls the risk gap. *Practical takeaway.* Single-model self-distillation stabilizes training and mitigates forgetting.

### A.3.5 Training Objective as a Joint-Likelihood Lower Bound

Our loss in Sec. 4 supervises $(P_t, P_v)$, $C$, and $a$. It can be seen as maximizing a lower bound on $\log p_\theta(a^\star \mid I, Q)$ marginalized over feasible traces.

**Proposition 4** (ELBO-style lower bound with feasible traces). *Let $\mathcal{T}_{\kappa,\rho}$ be the feasible set. For any auxiliary distribution $q(T \mid I, Q)$ supported on $\mathcal{T}_{\kappa,\rho}$,*

$$\log p_\theta(a^\star \mid I, Q) \geq \underbrace{\mathbb{E}_q[\log p_\theta(P_t, P_v \mid I, Q)]}_{\text{path term}} + \underbrace{\mathbb{E}_q[\log p_\theta(C \mid I, Q, P_t, P_v)]}_{\text{explanation term}}$$
$$+ \underbrace{\mathbb{E}_q[\log p_\theta(a^\star \mid I, Q, T)]}_{\text{answer term}} - \text{KL}(q(T \mid I, Q) \| p_\theta(T \mid I, Q, a^\star)). \tag{16}$$

*Proof.* Write $\log p_\theta(a^\star \mid I, Q) = \log \sum_{T \in \mathcal{T}_{\kappa,\rho}} p_\theta(T, a^\star \mid I, Q)$, insert $q(T \mid I, Q)$, and apply Jensen:

$$\log \sum_T q(T) \frac{p_\theta(T, a^\star)}{q(T)} \geq \mathbb{E}_q\big[\log p_\theta(T, a^\star) - \log q(T)\big].$$

Factorize $p_\theta(T, a^\star)$ using the model and rearrange. $\qquad\square$

*Intuition.* Supervising paths, explanations, and answers maximizes a tractable surrogate of the marginal likelihood; better selection of $q$ (stronger traces) tightens the bound. *Practical takeaway.* Improving the selector/feasibility checks translates into better training signals.

### A.3.6   PUTTING PIECES TOGETHER

Theorems 1–3 and Prop. 4 jointly suggest: (i) typed, verifiable traces reduce effective hypothesis space; (ii) the single-model selector is equivalent to MLE/MAP under a simple consistency–noise view; (iii) single-model self-distillation reduces supervision–generation shift; and (iv) the training objective maximizes a joint-likelihood lower bound whose tightness benefits from stronger traces and selection.

### A.4   USE OF LARGE LANGUAGE MODELS

In preparing this article, Large Language Models (LLMs) were employed only for stylistic refinement. Their role was limited to editing the wording of certain sections in order to improve readability and fluency of the manuscript. The intellectual contributions—including the development of ideas, design of experiments, analysis of results, and formulation of conclusions—were carried out entirely by the authors. No part of the research process, data interpretation, or scientific claims relied on the use of LLMs. The authors assume full responsibility for the content presented and ensure its originality and accuracy.

### A.5   DATA ETHICS STATEMENT

To evaluate the efficacy of StaR-KVQA, we conducted experiments which only use publicly available datasets, namely, OK-VQA (Marino et al., 2019) and FVQA (Wang et al., 2016). We also confirm that no personally identifiable information was utilized, and this research did not involve any human or animal subjects.

### A.6   PROMPTS

Following the methodology of ROG (Luo et al., 2023), we process the reasoning paths in two stages. First, we serialize each path by separating its constituent steps with a `<SEP>` token and terminating the sequence with `</PATH>`. Second, these serialized paths are parsed and converted into a structured format where consecutive steps are linked by an arrow ($\rightarrow$), e.g., `dog.color` $\rightarrow$ `dog.size`.

### A.7   QUALITATIVE CASE STUDY

In this section, we provide more qualitative case study examples.

**Prompt of Dual-Path Planner**

```
Given the image and the question below, generate exactly two relation paths
to help answer the question using a knowledge graph reasoning approach.
Follow the instructions precisely:
1. **vision_path**: Infer a visual reasoning path based on detectable
objects, scenes, or attributes in the image.
2. **text_path**: Derive a semantic reasoning path from the meaning of the
question using background knowledge.

IMPORTANT:
- Output ONLY two lines.
- Use the exact format:
 vision_path: <PATH> relation1 <SEP> relation2 <SEP> ... </PATH>
 text_path: <PATH> relation1 <SEP> relation2 <SEP> ... </PATH>
- Do NOT include any explanations, additional text, or extra lines.
- Replace relations with appropriate knowledge graph predicates (e.g.,
object.type, sports.use).
- Use <SEP> to separate each step in the path.
- End each path with </PATH>.

Example:
Question: What sport can you use this for?
text_path: <PATH> sports.equipment <SEP> sports.name </PATH>
image_path: <PATH> vehicle.type <SEP> vehicle.brand <SEP> sports.use </PATH>

Now answer the following:
{{Image}} {{Question}}
```

Figure 4: **Prompt of Dual-Path Planner.**

**Prompt of Reasoning Composer**

```
Based on the vision reasoning path {{vision_path}} and the text reasoning path
{{text_path}} , analyze the image to answer the question: {{ Question}} .

Use both paths as your primary guide for reasoning. Apply the visual path to
identify relevant objects or features in the image, and interpret the text
path to understand the semantic relationship needed. Combine both to reach a
clear conclusion.

Do not say the instruction is unclear or incomplete. Assume the paths are
valid and sufficient. Avoid emojis, disclaimers, or speculative language. Be
factual, concise, and directly derive the answer from the two reasoning paths.
```

Figure 5: **Prompt of Reasoning Composer.**

**Prompt of Best-Triplet Selector**

```
You are a judging module that outputs exactly one lowercase letter from the
provided choices.
Think silently; do not write your reasoning.

Primary criteria:
1) Explanation→Answer entailment.
2) Path→Explanation alignment (C explicitly references vision_path / text_path).
3) Visual plausibility from Image.
4) Coherence and concision.
5) a_pred vs ground_truth correctness (normalized; allow clear synonyms).

Output constraints:
- Output exactly one character from {{choices_letters}} .
- No spaces/newlines/punctuation.

Image: {{ Image}} Question : {{ Question}}

{{ #each candidates as |cand idx|}}
{{ choices_letters[idx]}}:
text_path : {{ cand.text_path}}
vision_path : {{ cand.vision_path }}
C: {{ cand.C}}
a_pred: {{ cand.ans}}
{{ /each}}

ground_truth: {{ ground_truth}}
```

Figure 6: **Prompt of Best-Triplet Selector.**

Table 5: Qualitative case study example #1, in OKVQA

**Question:** Can you name the place where this sport is played?

**Answers:** [rodeo, rodeo, rodeo, rodeo, equestrian park, equestrian park, ireland, ireland, kentucky derby, kentucky derby]

**Answer from Qwen2.5-VL-7B**

The image depicts a person on a horse performing a show jumping or dressage-like jump, which is commonly seen in equestrian competitions. The setting appears to be a Paddock Box or an arena used for equestrian sports.

**Answer from Gemini 2.5 Pro**

Based on the image, the sport is being played in an equestrian arena or a show ring. These arenas, typically filled with sand or a specialized footing, are found at: * Equestrian centers * Showgrounds * Fairgrounds * Large stables The event appears to be a form of equestrian exhibition or competition, possibly classical dressage, given the horse's "airs above the ground" posture (a move called a levade or pesade).

**Answer from StaR-KVQA**

**vision path:** horse.type → horse.color → sports.use
**text path:**  sports.name → sports.location

The sport depicted in the image is dressage, which is a discipline of horse riding that emphasizes the harmony and precision between the horse and its rider. Dressage is typically played in equestrian centers, stadiums, or specialized arenas designed for horse shows and competitions. Given the context of the image, the place where this sport is played can be inferred as an equestrian center or a stadium specifically set up for dressage competitions. The presence of spectators and the structured environment further support this inference.

Therefore, the possible answers include: **rodeo**, stadium, race track, horse race course.

Table 6: Qualitative case study example #2, in OKVQA

**Question:** How strong was the wind?

**Answers:** [very, very, very, very, 30mph, 30mph, very strong, very strong, gusty, gusty]

| **Answer from Qwen2.5-VL-7B** |
|---|
| In the image, the numerous kites flying high above the beach indicate that the wind was strong enough to lift the kites into the sky. The way the kites are spread throughout the sky also suggests varying wind speeds, likely creating a current suitable for kite-surfing or flying other types of kites. The fact that the people are comfortably engaging with the beach environment implies moderate wind, making it suitable for flying kites. The position of the clouds and the overall state of the beach atmosphere suggests a breezy yet not overwhelming day perfect for leisure activities like kite-flying. |
| **Answer from Gemini 2.5 Pro** |
| Based on the visual evidence in the image, the wind appears to be moderate and steady. Here's the reasoning: 1. Numerous Kites: The sky is filled with many kites of various shapes and sizes, including large, elaborate ones (like parafoils and animal shapes). These types of kites require a consistent and reasonably strong wind to get and stay aloft. 2. Kites are Flying High: The kites are flying high and seem stable, not dipping or struggling. This indicates the wind is not just a light, intermittent puff but a steady breeze. 3. Beach Umbrellas are Up: Despite the wind being strong enough for kites, people still have their beach umbrellas up. This suggests the wind is not excessively strong or gusty, as that would likely knock the umbrellas over or make them difficult to manage. This combination of many successfully flying kites and stable beach umbrellas points to a "perfect kite-flying day," which typically has winds in the 10-20 mph (16-32 km/h) range—a moderate breeze. |
| **Answer from StaR-KVQA** |
| **vision path:** kite.flying → kite.type → weather.wind 
 **text path:** weather.wind.strength → weather.condition |
| The image shows a beach scene with numerous kites flying in the sky. The presence of kites suggests that there is a sufficient amount of wind to keep them aloft. However, the image does not provide specific information about the wind speed or strength. To determine the wind strength, one would typically need additional data such as wind speed measurements or a more detailed analysis of the kite's behavior in the air. In the context of the image, the kites are flying, which indicates that there is some wind present. However, the exact strength of the wind cannot be determined from the image alone. The wind strength could range from a light breeze 
 Therefore, the possible answers include: 20 mph,**very**,100mph,strong |

Table 7: Qualitative case study example #3, in OKVQA

**Question:** Where in canada is this?

**Answers:** [alberta, alberta, alberta, alberta, toronto, toronto, rocky mountain, rocky mountain, vancouver, vancouver]

**Answer from Gemini 2.5 Pro**

Based on the visual evidence, this photo was taken at the Whistler Blackcomb ski resort in British Columbia, Canada. Here are the clues that point to this location: * Canadian Flags: The prominent display of multiple Canadian flags is a signature feature at a scenic viewpoint on Whistler Mountain. * Mountain Scenery: The vast, snow-covered mountains with rugged peaks and wide-open bowls are characteristic of the scenery at Whistler Blackcomb. The specific peak visible through the clouds is likely Whistler Peak itself. * Viewing Area: This specific setup, with the flags lining a railing at a popular photo spot, is located near the Roundhouse Lodge on Whistler Mountain. It's a famous location for tourists and skiers to take in the panoramic views.

**Answer from Qwen2.5-VL-7B**

This image features a snowy mountain landscape in Canada, recognizable by the snow-covered peaks and the Canadian flags. Ski resorts are common in mountainous regions across Canada, and the presence of these flags suggests that this is likely a popular ski or snowboarding destination. However, without more specific landmarks or signage, it's not possible to pinpoint the exact location within Canada. Some well-known ski resorts in Canada include Whistler Blackcomb in British Columbia and Mont Tremblant in Quebec, which are both known for their Canadian flags and snowy terrain.

**Answer from StaR-KVQA$_{qwen}$**

**vision path:** flag.type → flag.country
**text path:** location.name → location.country

The image shows a snowy mountain landscape with Canadian flags, suggesting that the location is in Canada. The presence of Canadian flags and the snowy terrain are strong indicators that this is a ski resort in Canada. However, the specific location cannot be determined from the image alone without additional context or landmarks. The predicted paths and ground truth paths provided seem to be related to the analysis of the image and the text content, but they do not directly provide the name of the location. The paths are likely part of a machine learning model or a text analysis tool that is trying to identify the location based on the image and the text content.

Therefore, the possible answers include: montana,**alberta**,british columbia

**Answer from Llama-3.2-11B-Vision**

The location of this scene appears to be in British Columbia, Canada, given the presence of large snow-covered mountains and ski lifts. Key indicators of the location include: * The presence of Maple Leaf flags, which are commonly used in Canadian national parks. * The snow-covered terrain and ski lifts, suggesting a winter sports destination. This conclusion is based on the visual cues present in the image.

**Answer from StaR-KVQA$_{llama}$**

**vision path:** flag.color → flag.shape → location.country
**text path:** location.country → location.name

The image appears to be taken at the top of a ski resort in Whistler, British Columbia, Canada, as indicated by the Canadian flags and the snow-covered mountains in the background.

Therefore, the possible answers include: ontario,whistler,**alberta**,vancouver

**Answer from Gemma-3-12B**

Based on the image, this is Revelstoke Mountain Resort in British Columbia, Canada. Here's how I identified it: * Canadian Flags: The prominent display of Canadian flags is a good indicator that this is likely in Canada. * Mountain Scenery: The steep, snow-covered mountains are characteristic of the Canadian Rockies and other mountain ranges in BC. * Ski Resort Infrastructure: The lift, fenced area, and layout suggest a ski resort. * Logo: The small logo at the bottom right of the image says "Revelstoke Mountain Resort". You can find similar images of the same scene by searching for "Revelstoke Mountain Resort flags" on the internet.

**Answer from StaR-KVQA$_{gemma}$**

**vision path:** scene.environment → scene.location → country.name
**text path:** location.country → location.region

Here's my analysis based on the image and the predicted paths: Likely Location: The image most likely depicts Whistler, British Columbia, Canada. Reasoning: * Canadian Flags: The prominent display of Canadian flags is a strong indicator. * Mountainous Terrain: The snowy, mountainous landscape is characteristic of British Columbia. * Ski Resort Setting: The presence of skiers, ski lifts, and a groomed snow area strongly suggests a ski resort. Whistler is a world-renowned ski resort in British Columbia. * Signage: The sign in the image appears to say "Lost and Found," which is consistent with signage found at Whistler Blackcomb. Therefore, based on these visual cues, Whistler, British Columbia, is the most probable location.

Therefore, the possible answers include: whistler,**alberta**,british columbia,calgary

Table 8: Qualitative case study example #4, in OKVQA

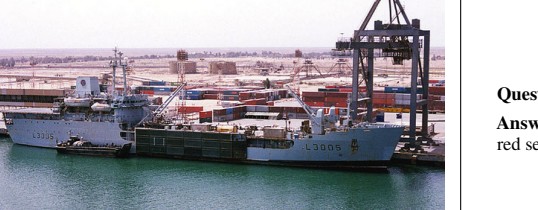

**Question:** Name the sea shown in this picture where the ship is seen?

**Answers:** [atlantic, atlantic, atlantic, atlantic, dock of red sea, dock of red sea, kamchatka, kamchatka, philippine, philippine]

**Answer from Gemini 2.5 Pro**

Based on the visual evidence, the ship in the image can be identified, which helps pinpoint the location. The ship with the pennant number L3005 is the British Royal Fleet Auxiliary landing ship RFA Sir Galahad. This photograph was taken at the port of Umm Qasr, Iraq, likely around 2003 during humanitarian relief operations following the coalition invasion. The port of Umm Qasr is situated on the Khawr Abd Allah waterway, which flows into the Persian Gulf. Therefore, the sea connected to the port shown in the picture is the Persian Gulf.

**Answer from Qwen2.5-VL-7B**

The image depicts a naval ship, specifically a Landing Ship Logistic (LSLV), which is docked at a port. The ships featured are likely British Royal Navy L Class landing ships, with the visible code "L3008" and "L3005" indicating specific vessels. These ships are designed to carry and transfer equipment, vehicles, and engineers required to deploy a boarding party and establish Befalton, a Mangonui, and Henry Field villages, often recognized in sailed stories. The port itself could be (100, Tenth City) Port located near Ascension Island, though Nigerian ports also encounter L Class ships, possibly in a military or logistical context. The exact location cannot be discerned without geographical features or more context.

**Answer from StaR-KVQA$_{qwen}$**

**vision path:** ship.type→ ship.location → sea.name
**text path:** ship.location → sea.name

The image shows a ship in a harbor with containers and industrial structures in the background. The ship appears to be a Royal Navy landing ship, as indicated by the markings "L3005" on the side. The location of the ship is not explicitly stated in the image, but based on the context and the style of the ship, it is likely in a port in the United Kingdom. The sea shown in the picture is the English Channel, which is a body of water separating the island of Great Britain from continental Europe. However, the image does not provide enough specific details to definitively identify the exact port or sea.

Therefore, the possible answers include: **atlantic**

**Answer from Llama-3.2-11B-Vision**

The location is not explicitly mentioned in the picture, but based on general context, the ship is likely at the Port of Cartagena or Port Oostende.

**Answer from StaR-KVQA$_{llama}$**

**vision path:** water.body → water.color → water.location
**text path:** sea.name → sea.location

The sea shown in this picture is the Mediterranean Sea.

Therefore, the possible answers include: pacific,**atlantic**,red sea

**Answer from Gemma-3-12B**

Here's a breakdown of the image and the answer to your question: 1. Object Identification and Attributes: * Ships: Two large, grey ships are prominently featured. They appear to be military or transport vessels, with the designation "L3005" visible on their hulls. * Containers: Numerous shipping containers in various colors (red, green, grey) are stacked near the ships and along the docks. * Dock/Port Area: A large port or dock area is visible, with cranes, storage areas, and other infrastructure. * Crane: A large gantry crane is present, likely used for loading and unloading cargo. * Water: A body of water (sea or harbor) surrounds the ships and docks. * Landscape: A flat, arid landscape is visible in the background. * Weather: The sky is hazy and the lighting suggests a bright, sunny day. 2. Relationship Path & Reasoning: The question asks "Name the sea shown in this picture where the ship is seen?". To answer this, we need to consider the context of the image. The presence of a large port, military-style ships, and the arid landscape provide clues. Based on the visual cues, this is likely the Persian Gulf. The arid landscape, the presence of a large port facility, and the type of ships are all consistent with the Persian Gulf region. 3. Answer: The sea shown in this picture is the Persian Gulf.

**Answer from StaR-KVQA$_{gemma}$**

**vision path:** scene.environment → scene.body of water → body of water.name
**text path:** location.type → location.name

Here's how we can approach this: 1. Identify the ship: The ship in the image is a Royal Fleet Auxiliary (RFA) ship, specifically an LST (Landing Ship Tank). The hull number "L3005" is visible. This identifies it as RFA Sir Galahad. 2. Determine the location: RFA Sir Galahad was involved in the Falklands War. It was sunk during the landings at Fitzroy Bay on the Falkland Islands. 3. Identify the sea: The Falkland Islands are located in the South Atlantic Ocean. Therefore, the sea shown in the picture is the South Atlantic Ocean.

Therefore, the possible answers include: **atlantic**,pacific

Table 9: Qualitative case study example #5, in FVQA

**Question:** Which object in this image is used for play?
**Answers:** sofa

| **Answer from Gemini 2.5 Pro** |
| --- |
| Based on the image provided, there are no explicit toys or games visible. However, the object most likely to be used for play, especially imaginative play for a child, is the rug. The rug features a pattern of stylized, repeating figures (which look like birds or other animals). Children often use such patterns on floors or rugs as a landscape for their games, moving toys around them or creating stories about the figures. |
| **Answer from Qwen2.5-VL-7B** |
| The image shows a living room with a couch, a rug, a floor lamp, and a wall decoration. None of these objects appear to be designed for play. The couch is for seating and relaxation, the rug is for decoration and possibly for warmth, the floor lamp is for illuminating the space, and the wall decoration is for decor. If you meant to identify an object intended for play, it might be the rug, which can sometimes be used for coffee table games and activities, or the floor, which could be used for games if a play mat or rug is used. However, there is no specific object in this image that is designed or intended for play in the sense of a toy, game, or sports equipment. |
| **Answer from StaR-KVQA**$_{qwen}$ |
| **vision path:** furniture.type→ furniture.use |
| **text path:** object.type → object.use |
| The image provided does not contain any objects that are typically used for play. The image shows a modern living room with a sofa, a rug, a lamp, and a piece of wall art. There are no toys, games, or objects that are commonly associated with play. Therefore, none of the objects in this image are used for play. |
| Therefore, the possible answers include: **sofa** |
| **Answer from Llama-3.2-11B-Vision** |
| Upon carefully examining the image provided, the carpet with a pattern of multicolored frogs appears to be the closest object tied to the act of play. The presence of the pattern suggests it's a fun and engaging element that could easily encourage playfulness. |
| **Answer from StaR-KVQA**$_{llama}$ |
| **vision path:** sofa.type → sofa.use |
| **text path:** furniture.type → furniture.use |
| he object used for play in this image is the couch. |
| Therefore, the possible answers include: **sofa** |
| **Answer from Gemma-3-12B** |
| Based on the image, the object most likely used for play is the rug. Here's why: * Location: It's placed on the floor in a living room setting, a common area for play. * Pattern: The rug has a playful, abstract pattern that could be engaging for children. * Function: Rugs are often used as play surfaces for kids. Let me know if you'd like me to analyze the image for other details! |
| **Answer from StaR-KVQA**$_{gemma}$ |
| **vision path:** people.group → people.activity → meditation.practice |
| **text path:** person.activity → meditation.practice |
| Based on the image, it's difficult to definitively say which object is used for play. However, the sofa is the most likely candidate. Sofas are often used for lounging, relaxing, and can be a place where children play. Here's why other objects are less likely: * Rug: Primarily a floor covering. * Floor lamp: A lighting fixture. * Artwork: Decorative. * Glass railing: A structural element. Therefore, the sofa is the most plausible object used for play in this image. |
| Therefore, the possible answers include: **sofa** |

