# OpenReview forum: "StaR-KVQA: Structured Reasoning Traces for Implicit-Knowledge Visual Question Answering"
_ICLR.cc/2026/Conference — ICLR 2026 Conference Withdrawn Submission_

### Official Review · Reviewer_euXg · 2025-10-28

**Soundness:** 2
**Presentation:** 2
**Contribution:** 2
**Rating:** 4
**Confidence:** 4

**Summary:**

This paper proposes StaR-KVQA (Structured Reasoning Traces for IK-KVQA), designed to address key challenges in Implicit-Knowledge Visual Question Answering (IK-KVQA) — namely, the lack of explicit reasoning supervision, weak interpretability, and poor cross-domain generalization.
To this end, StaR-KVQA introduces three core mechanisms:

1.Structured reasoning traces, composed of dual-channel symbolic relational paths (text path + vision path) and path-supported natural language explanations;

2.A unified model pipeline, where a single open-source MLLM (e.g., Qwen2.5-VL) is used to generate reasoning paths, craft explanations, and select the most consistent triplets;

3.Structured self-distillation, which fine-tunes the model on reasoning-trace-augmented datasets to enable learning of explicit reasoning processes.

StaR-KVQA achieves significant improvements on the OK-VQA and FVQA benchmarks, demonstrating strong cross-domain generalization and interpretability.

**Strengths:**

(1) StaR-KVQA introduces a novel perspective in the field of Implicit-Knowledge Visual Question Answering (IK-KVQA) by employing Structured Reasoning Traces to provide explicit supervision over the reasoning process of implicit-knowledge models.

1.Methodologically, it is the first to integrate symbolic relational paths with natural language explanations, achieving self-supervised reasoning enhancement through a single-model self-distillation framework.

2.At the task level, it redefines the boundary of “verifiable reasoning” within IK-KVQA, endowing implicit-knowledge models with explicit reasoning capabilities.

(2) StaR-KVQA holds notable theoretical and practical impact potential, providing a promising pathway toward verifiable reasoning in Implicit-Knowledge Visual Question Answering (IK-KVQA).

**Weaknesses:**

(1) In the abstract, the current wording may give the impression that StaR–KVQA is the first method designed for IK-KVQA, which is not accurate. A more appropriate approach would be to first summarize the limitations of existing IK-KVQA methods, then introduce StaR–KVQA as a method that specifically addresses these shortcomings.

(2) In the introduction and related work sections, there is a lack of in-depth analysis of existing IK-KVQA approaches. For instance, since StaR–KVQA is proposed to overcome the weaknesses of MLLM-based methods, it would be helpful to clarify whether current IK-KVQA methods merely rely on MLLMs for reasoning. This point requires further explanation and a more detailed discussion of the specific limitations of prior work.

(3) The notation used throughout the paper also shows some inconsistencies. For example, the symbol K is used both to denote external knowledge and the number of candidate path pairs, while in Equation (5), b suddenly replaces k without any explanation. Consistency and clarity in symbol usage should be improved.

(4) In Figure 2 (e), it remains unclear whether StaR–KVQA still requires fine-tuning of MLLMs during the reasoning stage. This design choice seems questionable and should be further clarified or justified.

(5) In Section 4.1 (DUAL-PATH PLANNER) and the hyperparameter analysis, the authors only investigate the impact of the number of generated dual paths (K) on performance, but they do not discuss whether there are explicit constraints on the length of each path or how such constraints might affect reasoning quality and efficiency.

(6) In StaR–KVQA, all reasoning paths and explanations are generated and evaluated by the same MLLM, which introduces a potential issue of “self-generation–self-verification” bias. Without external constraints or validation signals, the model might favor its own generation style rather than ensuring true logical consistency. For instance, if the initial reasoning paths contain errors, the model could self-distill on flawed reasoning patterns, leading to pseudo-consistency rather than genuine interpretability.

(7) In Table 3, the ablation variant (1) removes both the visual and textual reasoning paths, meaning that the reasoning composer and best-triplet selector should no longer function as intended—essentially reducing the system to a standard MLLM-based IK-KVQA. However, the results still show a notable performance improvement, which warrants further clarification. It would be helpful to explain what specific components contribute to this gain.

(8) Moreover, since the proposed method introduces additional stages—such as dual relational path generation and structured self-distillation—the experimental section should include a comparative analysis of computational cost (e.g., training time, inference latency, and memory usage) against other IK-KVQA baselines to assess practical efficiency and scalability.

**Questions:**

(1) In the abstract, the authors should first summarize the limitations of existing IK-KVQA methods, then introduce StaR–KVQA as a targeted solution that addresses these deficiencies. Additionally, the paper should provide a more detailed analysis of prior IK-KVQA approaches, clearly outlining their strengths and weaknesses, and explaining how StaR–KVQA improves upon these limitations.

(2) The notation throughout the paper should be standardized, with each symbol’s meaning clearly defined upon first appearance. The same symbol should not be used to represent different concepts, and all notations in the formulas should remain consistent with earlier definitions.

(3) The authors should also clarify whether StaR–KVQA requires fine-tuning of MLLMs during the reasoning stage, to prevent reader confusion. Furthermore, they should include an analysis of how reasoning path length affects model performance, specifying whether there are explicit constraints or optimal length ranges, to strengthen the evaluation of path design choices.

(4) The paper should further discuss the potential bias of the “self-generation–self-verification” mechanism, considering the introduction of external validation or constraints, or alternatively, provide empirical evidence demonstrating whether self-distillation leads to pseudo-consistency. Finally, it is recommended to include a comparative analysis of computational efficiency—such as inference time and memory usage—between StaR–KVQA and other IK-KVQA baselines, in order to assess the method’s practical efficiency and scalability.

---

### Official Review · Reviewer_J951 · 2025-10-30

**Soundness:** 2
**Presentation:** 2
**Contribution:** 2
**Rating:** 4
**Confidence:** 4

**Summary:**

This paper proposes StaR-KVQA, an IK-KVQA framework that trains MLLMs with structured reasoning traces: a pair of dual relation paths (text path + vision path) plus a path-grounded explanation, selected by an internal LLM-as-judge, and then fine-tuned end-to-end so inference is a single pass without retrieval. Reported gains are large on OK-VQA and FVQA, with claims of improved cross-domain generalization and interpretability. Several ablations on each components and number of traces are reported to justify the effectiveness of the method.

**Strengths:**

1. **Single-pass, IK-compliant pipeline.** The planner–composer–selector design keeps everything inside the in-knowledge (IK) setting and produces answer + reasoning trace in one shot, avoiding external retrieval and multi-stage brittleness.


2. **Large accuracy gains.** The method delivers strong improvements on standard IK-KVQA benchmarks, with ablations showing that each component (structured paths, rationale, selector) contributes meaningfully rather than the effect coming from one trick.


3. **Auditable reasoning traces.** The dual path (text + vision) plus path-grounded rationale yields step-by-step explanations that are inspectable and verifiable, improving trust and error analysis compared to free-form rationales.

**Weaknesses:**

1. **Motivation, definition and validation of “relation paths” is unclear.**

The paper leans on symbolic paths but motivation on using relational path needs further justified, as MLLM itself is not excessively trained on structured paths dataset, it remains a question why MLLMs can effectively extract the visual and textual attributes from the input image-question pair. The author should have discussed what is the ontology/label space for relations; how vision-side attributes (e.g., dog.color → dog.size) are extracted/validated, and how faithfulness of explanations to paths is measured beyond the selector’s preference.

2. **Interpretability evaluation is thin.**

The paper claims the proposed method improved explainability, however only accuracy is reported. A paper about faithful, verifiable reasoning should include explanation faithfulness metrics (e.g., path-coverage, counterfactual sensitivity, sufficiency/necessity tests) and human studies with model-annotator agreement.

3. **Cross-domain generalization is overclaimed.**

While the generated reasoning path on OK-VQA can improve the performance on FVQA and vice versa, these two dataset innately share the similar distribution of question and image types on real-world knowledge-based VQA task. Experiments on training STAR-KVQA on general KVQA datasets and test it on other domains of KVQA dataset like MMMU (reasoning and expert knowledge), ChartQA (diagram understanding), VSR (spatial reasoning) can further demonstrate the generalizability.

**Questions:**

What is the relation inventory (size, examples, typing)? How are paths parsed/validated, and how often do paths include relations not visually supportable?

Beyond accuracy, how do you quantify that explanations of the generated paths (coverage, alignment, contradiction rate)?

Can you demonstrate the generalizability of the proposed method on other OOD benchmarks like MMMU? VCR? VSR? ChartQA? MathVista?

[1] ChartQA: A Benchmark for Question Answering about Charts with Visual and Logical Reasoning

[2] From Recognition to Cognition: Visual Commonsense Reasoning

[3] MMMU: A Massive Multi-discipline Multimodal Understanding and Reasoning Benchmark for Expert AGI

[4] MathVista: Evaluating Mathematical Reasoning of Foundation Models in Visual Contexts

[5] VSR: Visual Spatial Reasoning

---

### Official Review · Reviewer_LAdB · 2025-11-01

**Soundness:** 2
**Presentation:** 3
**Contribution:** 3
**Rating:** 6
**Confidence:** 3

**Summary:**

This paper studies the implicit-knowledge KVQA (IK-KVQA) task, which differs from traditional KVQA by disallowing access to external knowledge bases. In this setting, vision-language models must answer knowledge-intensive questions relying solely on their internal, parametric knowledge. This setup reflects realistic use cases where off-the-shelf VLMs must reason beyond purely in-context information from the input image and text.
The proposed method, StaR-KVQA, bootstraps a pretrained VLM’s internal parametric knowledge to construct structured reasoning traces and finetunes the model using these traces. The approach achieves substantial gains over both previous knowledge-based systems and proprietary VLMs such as GPT-4o and Gemini 2.5 Pro. The paper also includes ablations and cross-domain generalization experiments (OK-VQA <-> FVQA), demonstrating the method’s effectiveness and robustness.

**Strengths:**

- The IK-KVQA task formulation is well motivated, reflecting practical deployment scenarios where retrieval is unavailable.
- The proposed structured self-distillation approach is novel and technically clean, using only a single model family throughout planning, reasoning, selection and finetuning.
- The inclusion of ablation studies and cross-domain generalization (between OK-VQA and FVQA) further supports the effectiveness and generalizability of their proposed approach.

**Weaknesses:**

- The benefit of the dual-path planner (text vs. vision relation paths) over a more generic chain-of-thought style fine-tuning is unclear. For example, in the motivating case “Which breed of dog is this?”, relations such as dog.color → dog.size or dog.breed → dog.name do not evidently support the final answer, and this decomposition seems somewhat artificial/unnecessary.
- A direct baseline using prompt optimization to induce CoT reasoning is not provided and could be informative. If a well-optimized CoT-prompted fine-tuning achieves comparable performance, the claimed advantage of the dual-path structure may weaken. Including such comparisons to CoT + SFT would better isolate the contribution of the structured-trace supervision.
- The authors could further discuss the faithfulness of the generated reasoning traces to strengthen their interpretability claim. In particular, it remains unclear whether these traces genuinely guide the model’s answer generation, or instead function primarily as auxiliary token sequences that facilitate latent reasoning without reflecting the actual decision process.

**Questions:**

- Could the authors report CoT-prompted results for the strongest closed-source models (e.g., GPT-4o, Gemini 2.5 Pro)? Would StaR-KVQA still outperform them under comparable prompting?
- (Minor) In lines 318–319 and 368, does “COT” refer to chain of thought? If so, please standardize to “CoT” to avoid confusion.
- (Minor) In Table 3, clearer names for the five ablation variants (e.g., No-Paths, No-Composer, text-only, vision-only, No-Selector, etc.) would improve readability.

---

### Note · Authors · 2025-11-12

I have read and agree with the venue's withdrawal policy on behalf of myself and my co-authors.